# Epidemiological Study of Multiple Zoonotic Mosquito-Borne Alphaviruses in Horses in Queensland, Australia (2018–2020)

**DOI:** 10.3390/v14091846

**Published:** 2022-08-23

**Authors:** Ka Y. Yuen, Joerg Henning, Melodie D. Eng, Althea S. W. Wang, Martin F. Lenz, Karen M. Caldwell, Mitchell P. Coyle, Helle Bielefeldt-Ohmann

**Affiliations:** 1School of Veterinary Science, The University of Queensland, Gatton, QLD 4343, Australia; 2Queensland Racing Integrity Commission, Brisbane, QLD 4010, Australia; 3Equine Unit, Office of the Director Gatton Campus, The University of Queensland, Gatton, QLD 4343, Australia; 4School of Chemistry and Molecular Biosciences, The University of Queensland, Brisbane, QLD 4072, Australia; 5Australian Infectious Diseases Research Centre, The University of Queensland, Brisbane, QLD 4072, Australia

**Keywords:** Ross River virus, Sindbis virus, Barmah Forest virus, alphavirus, spatial analysis, risk factors, seroprevalence, infectious diseases, cross-reactivity

## Abstract

The increased frequency of extreme weather events due to climate change has complicated the epidemiological pattern of mosquito-borne diseases, as the host and vector dynamics shift to adapt. However, little is known about the seroprevalence of common mosquito-borne virus infections in horses in Australia. In this study, serological surveys for multiple alphaviruses were performed on samples taken from 622 horses across two horse populations (racehorses and horses residing on The University of Queensland (UQ) campus) in Queensland using the gold standard virus neutralization test. As is the case in humans across Australia, Ross River virus (RRV) is the most common arbovirus infection in horses, followed by Barmah Forest virus, with an overall apparent seroprevalence of 48.6% (302/622) and 4.3% (26/607), respectively. Horses aged over 6 years old (OR 1.86, *p* = 0.01) and residing at UQ (OR 5.8, *p* < 0.001) were significantly associated with seroconversion to RRV. A significant medium correlation (r = 0.626, *p* < 0.001) between RRV and Getah virus (GETV) neutralizing antibody titers was identified. Collectively, these results advance the current epidemiological knowledge of arbovirus exposure in a susceptible host in Australia. The potential use of horses as sentinels for arbovirus monitoring should be considered. Furthermore, since GETV is currently exotic to Australia, antibodies cross-reactivity between RRV and GETV should be further investigated for cross-protection, which may also help to inform vaccine developments.

## 1. Introduction

Mosquito-borne diseases are highly dynamic in their transmission patterns, especially in the era of climate change. In Australia, Ross River virus (RRV) and Barmah Forest virus (BFV), in the genus Alphavirus, family *Togaviridae*, are of concern to both the medical and veterinary professions. Transmission cycles of alphaviruses are very similar, in a reservoir-mosquito-susceptible host fashion. Generally, the main reservoir hosts for arthropod-borne alphaviruses in Australia are marsupials, although other wildlife species have also been implicated [1]. Many mosquito species within the genus *Aedes* and *Culex* are competent in transmitting alphaviruses [2,3]. Both horses and humans are susceptible to infection by several alphaviruses, displaying a wide range of clinical presentations, ranging from asymptomatic, to neurological and chronic musculoskeletal signs (Table 1).

Among all arboviruses endemic to Australia, RRV is the most economically important and the most commonly implicated in human infection [1,3]. Considerable research efforts have focused on the epidemiology of RRV in kangaroos (reservoir hosts) and mosquitoes (vectors), as well as on the development of predictive modeling [2,4]. However, there appears to have been little advancement in the knowledge of the disease dynamics of RRV in recent years. Despite horses being arguably the companion animal most susceptible to a variety of arboviruses [5,6,7,8,9,10], little is known about the seroprevalence of these viruses in horses. In contrast to human medicine, RRV infection in horses is not a notifiable disease. The last Australia-wide (excluding Tasmania) seroprevalence survey for RRV in horses was performed in 2011, and it identified a 21% seropositivity rate amongst approximately 1000 horses presenting with neurological signs [7]. The most recent state-specific surveys identified a 91% seropositivity rate in north Queensland in 2014 [11]; a 56% seropositivity rate in Victoria in 2000–2002 [12]; and a 62% seropositivity rate in New South Wales (NSW) in 1982–1983 [13]. Barmah Forest virus (BFV) is the second most common notifiable alphavirus infection in humans in Australia [14], but little is known about its prevalence in horses. The prevalence of Sindbis virus (SINV) remains unknown in Australia, as infection is usually considered to cause only mild symptoms and is non-notifiable in both human and veterinary medicine [15,16]. However, recent studies from South Africa reported the ability of SINV to cause a fatal neurological disease in naturally infected humans and horses [17,18]. Although Getah virus (GETV) is not zoonotic and is exotic to Australia, the high level of antibody cross-reactivity with RRV [19] could affect exotic disease surveillance efforts in Australia. While clinical signs of infected horses are mild and non-life-threatening [20], incursion of GETV would pose a biosecurity risk to piggeries in Australia, as infection can cause neurological signs, abortion storm, and high mortality [21,22]. Outbreaks of GETV infections have also recently re-emerged in racehorses in Japan [9,10].

The objectives of this study were to determine, in relation to horses in Queensland, (1) the seroprevalence of various alphaviruses (RRV, BFV, SINV), (2) the neutralizing antibody cross-reactivity to related viruses within the same serocomplex group (specifically between RRV and the exotic GETV [19]), and (3) the spatial and risk factors associated with RRV infections.

## 2. Materials and Methods

### 2.1. Sample Collection

Animal ethics approval was obtained from The University of Queensland (UQ) Production Animal Ethics Committee (permit number SVS/344/18) prior to commencement of the study. Over a three-year period (2018–2020), a total of 732 serum samples were obtained from two categories of horses: (1) racehorses attending race meetings in QLD (*n* = 579); and (2) horses residing at UQ (Pinjarra Hills and Gatton; *n* = 153). Serum samples collected from racehorses were provided by the Queensland Racing Integrity Commission (QRIC) as an aliquot of samples taken for race-day prohibited-substance testing, then transported on ice and stored at −20 ºC upon receival by UQ. A total of 5 mL of blood was collected from UQ horses via jugular venipuncture with a 22-gauge needle into a Vacutainer^®^ containing clot activator. The blood samples were allowed to clot at room temperature for 1 h prior to centrifugation at 1342× *g* (2000 rpm), at 20 °C, for 20 min. Serum was aliquoted into sterile vials and stored at −20 °C until analysis. All serum samples were heat inactivated at 56 °C for 30 min prior to laboratory testing.

### 2.2. Cell Culture and Virus Stock Production

Vero cells (African green monkey-(*Chlorocebus* sp.) derived kidney epithelial cells) and C6/36 cells (*Aedes albopictus* larvae mid-gut cells) were cultured as previously described [23]. Virus seed stocks used in this study were: RRV (Cairns 2007; [24]), BFV (isolate 2193), SINV (isolate 23588), and GETV (isolate 1996). C6/36 cells were used for virus stock amplification. Briefly, 5–10 μL of the virus seed stock was used to infect a 85% confluent monolayer of C6/36 cells, cultured in Roswell Park memorial Institute (RPMI) medium (Gibco, Life Technologies Corporation, Carlsbad, CA, USA), supplemented with penicillin, streptomycin, L-glutamine (PSG; Gibco, Life Technologies Corporation, USA), in a 175 cm^2^ culture flask with a non-vented cap (Greiner Bio-One, Frickenhausen, Germany). After incubation at 28 °C for 1–2 h with intermittent rocking for virus attachment, the inoculum was removed and replace with RPMI medium supplemented with 2% fetal bovine serum (FBS; Gibco, Life Technologies Corporation, USA) and PSG. Amplified virus stock was harvested on day 4. Supernatant was collected and clarified by centrifugation at 4 °C at 1500× *g* in a 50 mL tube (Corning, Corning Incorporated, New York, NY, USA), then FBS concentrated and adjusted to 10%. Virus stocks were snap frozen and stored at −80 °C in 1 mL aliquots. Virus titer was determined by virus titration assay, as described previously [23], and calculated as TCID_50_ infectious units/mL.

### 2.3. Virus Neutralisation Test (VNT)

Each sample was tested for the presence of neutralizing antibodies for the various viruses in duplicate, initially serially diluted from 1:10 to 1:80 in Dulbecco’s modified eagle medium (DMEM; Gibco, Life Technologies Corporation, USA) supplemented with PSG, and 2% FBS in a 96-well micro-titer plate (Costar, Corning Incorporated-Life Sciences, Shanghai, China). A total of 100 infectious units of the virus in 50 μL of medium was added into each well to yield a final serum dilution of 1:20 to 1:160, and incubated at 37 °C with 5% CO_2_ humidified air for 1 h. Vero cells were seeded at 2 × 10^4^ cells per well in 50 μL of medium to give a final volume of 150 μL per well. The cytopathic effect (CPE) was examined and antibody titer level was determined after 5 days of incubation. A positive and negative control was included in each plate. A back titration was set up for each batch of testing to ensure that the results between batches tested on different days were comparable.

### 2.4. Statistical Analysis

The dataset was finalized using Microsoft Excel Spreadsheet. The following exclusion criteria were applied prior to statistical analysis: (1) racehorse training location outside of Queensland (QRIC, *n* = 50), or not known/provided (QRIC, *n* = 14); (2) less than 2 years old (UQ, *n* = 41); (3) toxic or inadequate serum samples (UQ, *n* = 2; QRIC, *n* = 1); (4) signalment information not known/provided (UQ, *n* = 2). The final dataset for analysis included horse age, breed, gender, location by suburb (training location for QRIC racehorses), as well as the sampling date for 622 horses. Locations were converted to their respective statistical local area level 3 (SLA 3), as defined by the Australian Bureau of Statistics [25].

Horses with antibody titer ≥ 1:20 in VNT were determined to be seropositive for the respective viruses (except GETV). Overall apparent prevalence (AP) was determined for all viruses. AP for RRV was determined for each SLA and visualized using QGIS (version 3.22). Cross-reactivity between GETV and RRV neutralizing antibodies were tested with Pearson’s correlation coefficient. For risk factor analysis, a 2-step approach was used. Univariant logistic regression analysis was used to identify risk factors, with factors that had significance values of *p* ≤ 0.2 being included in the subsequent multivariant logistic regression analysis model, with significance, in turn, being determined at *p* ≤ 0.05. Analyses were performed using STATA BE (version 17.0).

Spatial analysis was performed using R package spdep (version 1.2-2). A row-standardized weight list was first created prior to global and local auto-correlation analysis. For global spatial auto-correlation, Moran’s *I* statistics and *p*-values were calculated. For local spatial auto-correlation (or local indicator of spatial association (LISA)), local Moran’s *I* and *p*-values were calculated and results visualized using R package ggplot2 (version 3.3.5) in 4 categories: high-high, high-low, low-high, and low-low (Appendix A). Both global and local spatial auto-correlation analyses were performed in the entire dataset by seasons.

## 3. Results

### 3.1. Sample Population

Of the 622 samples included in this study, 82.6% (514/622) were from QRIC and 17.4% (108/622) from UQ, of which 42.6% (265/622) were female and 57.4% (357/622) were males. The mean age of the population was 5.8 years (SD = 3.7, range 2–24). The majority breed was Thoroughbred (85.4%, 531/622), followed by Standardbred (8.2%, 51/622), and Australian Stockhorse (4.0%, 25/622) (Table 2).

### 3.2. Serological Surveys of Alphaviruses

Heat-inactivated serum samples were tested for the presence of neutralizing antibodies for RRV, BFV, SINV, and GETV (Table 3). Overall, 48.6% (302/622) of the total study population were seropositive for RRV, of which 40.9% (210/514) of QRIC samples and 85.2% (92/108) of UQ samples tested positive. Overall AP of BFV and SINV were 4.3% (26/607) and 2.0% (12/611), respectively. Interestingly, there was a significant difference (*p* < 0.001) in BFV seroprevalence between the UQ (20.2%, 19/94) and QRIC populations (1.4%, 7/513), whereas there was no significant difference (*p* = 0.702) in SINV AP between the two populations (Table 3).

Of the 302 samples which tested positive to RRV, 281 samples were tested for neutralizing antibody cross-reactivity to GETV. The presence of cross-neutralizing antibody was detected in 77.6% (218/281) samples, with a significant medium correlation identified between antibody titer levels (r = 0.626, *p* < 0.001) (Figure 1).

### 3.3. Spatial Analysis of RRV Seroprevalence in QLD

When classified by their SLA 3, the majority (91.7%, 99/108) of UQ horses resided in the Lockyer Valley, accounting for 90% (99/110) of the study population in the Ipswich Hinterland. Within the UQ population, 85.2% (92/108) were seropositive for RRV, of which 90.2% (83/92) resided in Ipswich Hinterland, and the remaining (9.8%, 9/92) at the Kenmore–Brookfield–Moggill region (Table 4, Figure 2 and Figure 3).

For the racehorse population, most horses were trained at Nundah (33.1%, 170/514), with an AP of 35.3% (60/170), followed by the Gold Coast Hinterland (19.1%, 98/514), with an AP of 39.8% (39/98), then at the Sunshine Coast Hinterland (16.7%; 86/514), with an AP of 54.7% (47/86), and 14.8% (76/514) trained at Toowoomba, with an AP of 34.2% (26/76) (Table 4, Figure 2 and Figure 3).

The global spatial auto-correlation analyses of all datasets returned statistically non-significant results (*p* > 0.05; Table 5). Therefore, the null hypothesis, stating that RRV seropositivity rates in QLD were randomly distributed, was accepted. This also suggests that non-random clustering was not detected.

### 3.4. Risk Factors Analysis of RRV Seropositivity in Horses

The high seropositivity of RRV in both the QRIC and UQ horse populations prompted further analysis to better understand the risk factors associated with RRV infection in horses. In the univariant model, horses located at UQ (*p* < 0.001) and horses over 6 years old (*p* < 0.001) were more likely to test positive for RRV, whereas samples collected in summer (*p* = 0.045) and winter (*p* = 0.001) were less likely to test positive for RRV (Table 6). After multivariant adjustment, horses located at UQ were 5.8 times (*p* < 0.001) more likely to be seropositive for RRV compared to the QRIC horses, and horses aged over 6 years were 1.9 times (*p* = 0.01) more likely to be seropositive for RRV compared to 2–6-year-old horses (Table 6). The inclusion of season in the multivariant model returned a statistically non-significant Wald-test result (*p* = 0.380); therefore, season was excluded from the final multivariant model. Interactions between risk factors were also assessed in the final multivariant model; however, none were statistically significant (*p* > 0.05; Appendix A). The Akaike information criterion (AIC) of the final multivariant model was 785.3.

## 4. Discussion

This investigation represents an extensive serosurveillance study of mosquito-borne zoonotic alphaviruses in horses in Queensland, Australia. The results presented are of importance to veterinary medicine, and more broadly to the public health discipline, by furthering knowledge of the epidemiology of zoonotic arboviruses in a susceptible host, and identifying the risk factors associated with RRV infection in horses, in particular.

In line with statistics in human medicine in Australia [14,26,27,28,29], RRV is also the most prevalent arbovirus in horses, followed by BFV. This was most evident in the UQ horse population, where horses residing at the UQ campus in the Lockyer Valley were 6 times more likely to be seropositive for RRV than the general racehorse population in QLD. Within the 1060 hectares of the UQ Gatton campus, there are intense year-round farming systems of multiple animal species and agricultural practices [30]. Being located at a regional area with relatively little human traffic and constant feed availability, marsupials (wallabies and possums) are often found foraging in the paddocks where domestic animals are kept. Irrigation systems from the intense agricultural practices often lead to the build-up of stagnant water puddles in the paddocks suitable for mosquito breeding, especially during warmer months. This in itself allows the maintenance of RRV within the reservoir hosts and vector cycle. In addition, at any point in time, there are around 100 horses kept on campus. The high density of the horse population is likely to favor the transmission of RRV from infected mosquitoes to horses [31]. This is, however, different in the context of racehorses. While racehorses are often trained in one location, they frequently travel around the state, country, or overseas to compete. Moreover, racehorse training centers, in general, often have lower horse population densities. It is out of the scope of this study to assess whether agricultural practices contribute to the transmission of RRV in horses, as precise training locations were not provided due to client confidentiality. Nevertheless, proximity to agricultural practices being a risk factor cannot be ruled out.

In the top four statistical local areas where samples from racehorses were collected (representing 84% of the racehorse sample population), the Sunshine Coast Hinterland has the highest seroprevalence to RRV, and is the only SLA that has a seroprevalence above 50%. This is likely due to the combination of a high abundance of *Aedes vigilax* and wildlife populations, the presence of a wild horse population, and the agricultural practices in the Sunshine Coast Hinterland [32,33].

Many studies identified that high rainfall and seasonality were associated with increased RRV notification in humans (reviewed in [1]). While our study did not identify season as a risk factor of RRV infection in horses, this is likely due to the sampling bias in our study populations. The majority of the UQ population were sampled in spring, while the racehorse samples were scattered throughout the year (Appendix A). Further longitudinal studies in a naïve horse population are required to determine whether seasonality is a risk factor contributing to the transmission of RRV in horses.

Viruses within the same serogroup often have neutralizing antibodies that cross-react with each other as they share similar properties in their structural proteins [34]. As GETV remains exotic to Australia [19], the presence of neutralizing antibodies to GETV in this study indicates cross-reactivity between GETV and RRV neutralizing antibodies. This is of particular relevance for two important areas: (1) the ongoing effort of exotic disease surveillance, and (2) vaccine development, as vaccination against RRV might provide partial protection against GETV, and vice versa. While studies in mice demonstrated cross-protection between RRV and GETV [19], it is important to demonstrate reproducibility in naturally susceptible hosts, as upon experimental challenge, mice only partially display clinical signs experienced by the natural susceptible hosts [1]. It is also interesting to note the high seroprevalence of RRV compared to other alphaviruses, i.e., BFV and SINV, indicating that neutralizing antibodies to these viruses do not cross-react. While it is out of the scope of this study to determine the reasons for the differences in seroprevalence between the three viruses in horses, one could speculate that this may be due to virus competition within the mosquito vector, as demonstrated in flaviviruses [35]. However, other factors, such as vector competence, and viremia level and duration in the reservoir and susceptible hosts, should not be overlooked.

## 5. Conclusions

Collectively, results from this study further our knowledge of the current epidemiology of arboviruses in horses in Australia. The potential use of horses as sentinels of arbovirus monitoring to inform public health measures should be considered. Further, RRV transmission dynamics studies in horses could be performed within the UQ horse population, due to its high seroprevalence. The cross-reactivity between RRV and GETV neutralizing antibodies may inform future public health measures, should an outbreak of either virus occur. While it is out of the scope of this study to assess cross-protection against disease between RRV and GETV, further investigation is warranted to characterize and understand the underlying mechanisms of the observed cross-reactivity.

## Figures and Tables

**Figure 1 viruses-14-01846-f001:**
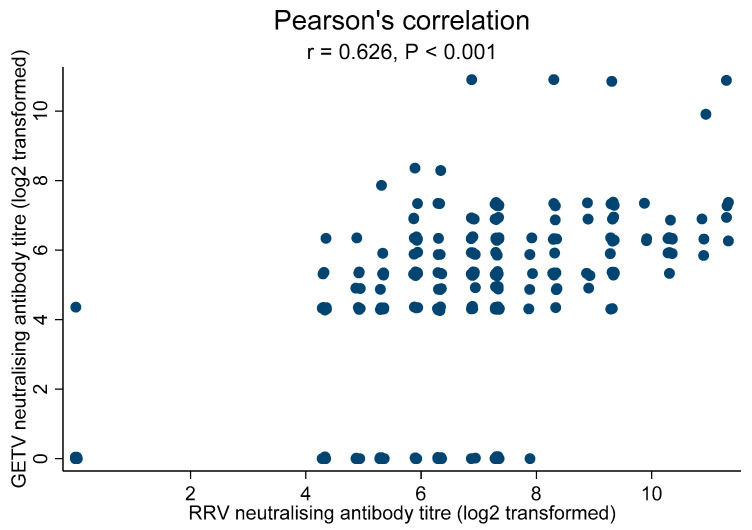
Correlation of neutralizing antibody titer (log_2_ transformed) between RRV and GETV. The random noise effect was added to the same data points to prevent overplotting.

**Figure 2 viruses-14-01846-f002:**
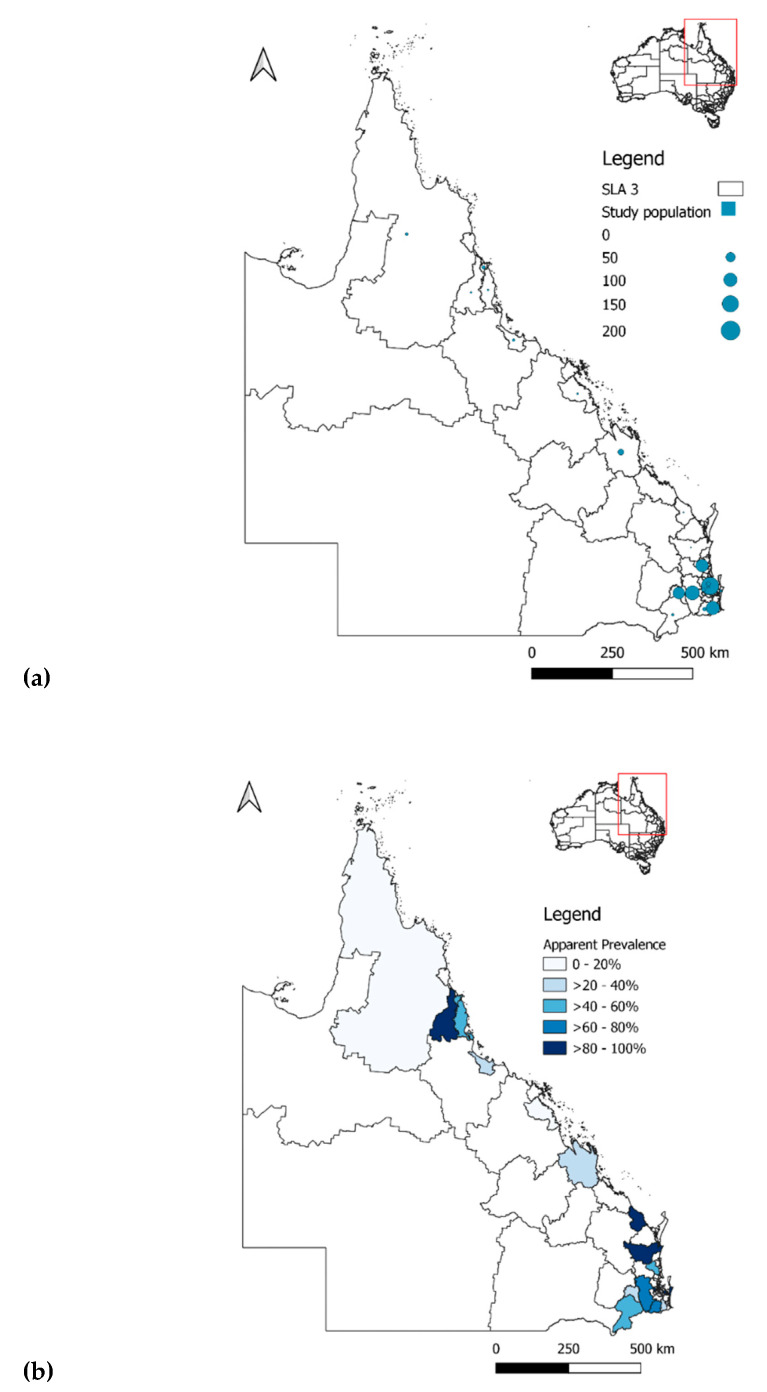
Maps of QLD classified by SLA 3 showing (**a**) distribution of sample population, and (**b**) apparent seroprevalence of RRV.

**Figure 3 viruses-14-01846-f003:**
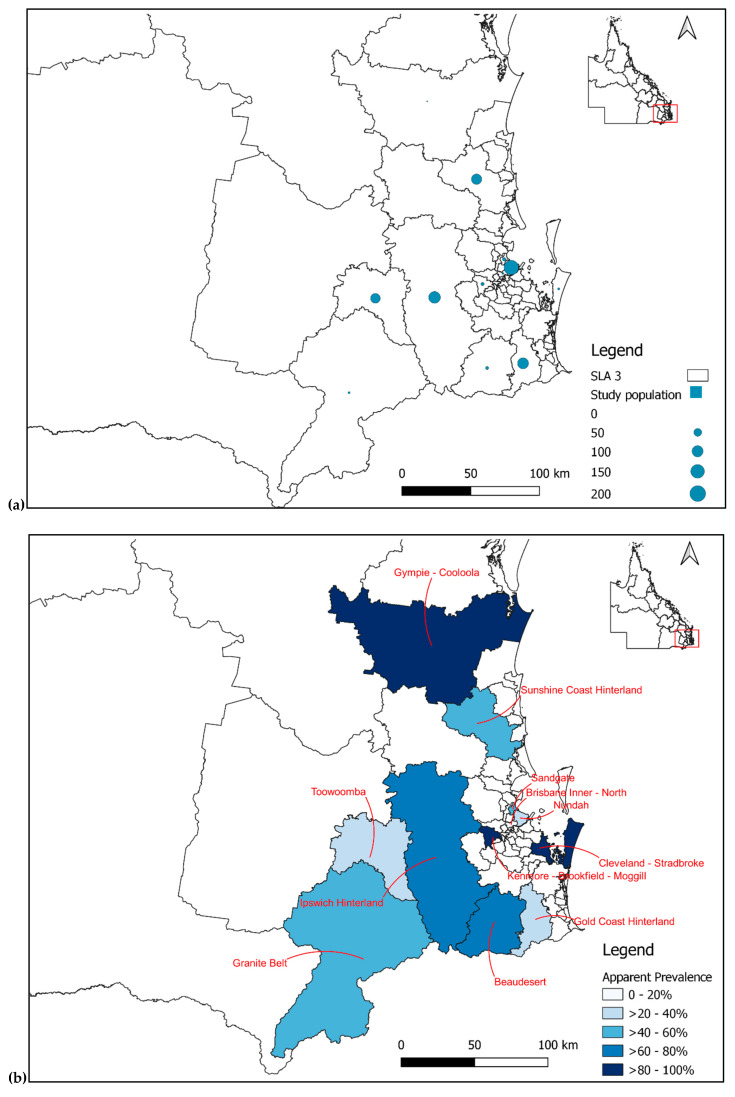
Maps of southeast QLD classified by SLA 3 showing (**a**) distribution of sample population; (**b**) apparent seroprevalence of RRV.

**Table 1 viruses-14-01846-t001:** Alphaviruses of concerns in Australia included in this study.

Examples	Zoonotic	Status in Australia	Type of Clinical Signs (Animals)
Ross River virus	Yes	Endemic	Musculoskeletal
Barmah Forest virus	Yes	Endemic	Musculoskeletal
Sindbis virus	Yes	Endemic	Musculoskeletal/Neurological
Getah virus	No	Exotic	Systemic

**Table 2 viruses-14-01846-t002:** Description of the sample population.

Category	Level	QRIC	UQ	Total
**Sex**	Male	318	39	357
	Female	196	69	265
**Age at sampling (years)**	2–6	447	27	474
	>6	67	81	148
**Breed**	Australian Stockhorse	0	25	25
	Australian Stockhorse × Standardbred	0	7	7
	Quarter horse	0	2	2
	Standardbred	0	51	51
	Thoroughbred	514	17	531
	Warmblood	0	6	6

**Table 3 viruses-14-01846-t003:** Outcome summary of alphaviruses serosurvey.

	RRV % (*n*)		BFV % (*n*)		SINV % (*n*)	
	Seropositive	Seronegative	Seropositive	Seronegative	Seropositive	Seronegative
**QRIC**	40.9 (210)	59.1 (304)	1.4 (7)	98.6 (506)	2.1 (11)	97.9 (503)
**UQ**	85.2 (92)	14.8 (16)	20.2 (19)	79.8 (75)	1.0 (1)	99.0 (96)
***p*-value ***	*p* < 0.001		*p* < 0.001		*p* = 0.702	

* Fisher’s exact test. RRV = Ross River virus; BFV = Barmah Forest virus; SINV = Sindbis virus.

**Table 4 viruses-14-01846-t004:** Distribution of QRIC and UQ sample population across QLD.

	QRIC		UQ	
Statistical Local Area 3 (SLA 3)	RRV Seropositive	RRV Seronegative	RRV Seropositive	RRV Seronegative
Cleveland–Stradbroke	4	0	0	0
Nundah	60	110	0	0
Sandgate	5	5	0	0
Kenmore–Brookfield–Moggill	0	0	9	0
Brisbane Inner–North	0	3	0	0
Cairns–South	4	3	0	0
Innisfail–Cassowary Coast	1	1	0	0
Tablelands (East)–Kuranda	2	0	0	0
Granite Belt	2	2	0	0
Rockhampton	8	12	0	0
Gold Coast Hinterland	39	59	0	0
Ipswich Hinterland	4	7	83	16
Beaudesert	5	3	0	0
Mackay	0	2	0	0
Far North	0	5	0	0
Sunshine Coast Hinterland	47	39	0	0
Toowoomba	26	50	0	0
Townsville	1	3	0	0
Bundaberg	1	0	0	0
Gympie–Cooloola	1	0	0	0
**Total**	**210**	**304**	**92**	**16**

**Table 5 viruses-14-01846-t005:** Global spatial autocorrelation (presented as Moran’s *I* statistics) of overall and season analyses for RRV seropositivity in QLD.

Dataset	Moran’s *I*	*p*-Value
Overall	–0.02734297	0.437
Spring	0.04030128	0.301
Summer	–0.1685299	0.361
Autumn	0.06044266	0.254
Winter	–0.11316486	0.549

**Table 6 viruses-14-01846-t006:** Risk factors analysis using logistic regression models.

				Univariant Analysis		Multivariant Analysis	
Risk Factors	Level	RRV Seropositive % (*n*)	RRV Seronegative % (*n*)	Odds Ratio (95% CI)	*p*-Value	Odds Ratio (95% CI)	*p*-Value
**Sex**	Female	48.3 (128)	51.7 (137)	1			
	Male	48.7 (174)	51.3 (183)	1.02 (0.74–1.40)	*p* = 0.914		
**Group**	QRIC	40.9 (210)	59.1 (304)	1		1	
	UQ	85.2 (92)	14.8 (16)	8.32 (4.76–14.56)	*p* < 0.001	5.84 (3.16–10.80)	*p* < 0.001
**Age at sampling (years)**	2–6	41.8 (195)	58.2 (271)	1		1	
	>6	72.3 (107)	27.7 (41)	3.73 (2.49–5.59)	*p* < 0.001	1.86 (1.16–2.97)	*p* = 0.010
**Season**	Spring	57.6 (95)	42.4 (70)	1	*p* = 0.004 *		
	Summer	45.9 (61)	54.1 (72)	0.62 (0.39–0.99)	*p* = 0.045		
	Autumn	54.4 (62)	45.6 (52)	0.88 (0.54–1.42)	*p* = 0.598		
	Winter	40.0 (84)	60.0 (126)	0.49 (0.32–0.74)	*p* = 0.001		

* Wald-test.

## Data Availability

Not applicable.

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
