# Peer review of "Epidemiological Study of Multiple Zoonotic Mosquito-Borne Alphaviruses in Horses in Queensland, Australia (2018–2020)"

_viruses, 2022, doi:10.3390/v14091846_

Round 1
Reviewer 1 Report
In this paper, Yuen et al describe the presence of antibodies against four Australian alphaviruses in horses in the state of Queensland, Australia. The manuscript is well written and enjoyable to read. The introduction is well presented, the methods and techniques are appropriate, the sample size is good too although the sample . Results are presented decently and conclusion is well founded on those results.
Related Old World alphaviruses can antigenically cross-react. The issue of antibody cross-reactivity between the four alphaviruses is discussed and, in my opinion, it does not affect the conclusions of the paper.
The last paragraph in the introduction, where the aims of the study are stated, needs to be more concise.
Author Response
Reviewer 1
In this paper, Yuen et al describe the presence of antibodies against four Australian alphaviruses in horses in the state of Queensland, Australia. The manuscript is well written and enjoyable to read. The introduction is well presented, the methods and techniques are appropriate, the sample size is good too although the sample . Results are presented decently and conclusion is well founded on those results.
We thank the reviewer for the positive comments.
Related Old World alphaviruses can antigenically cross-react. The issue of antibody cross-reactivity between the four alphaviruses is discussed and, in my opinion, it does not affect the conclusions of the paper.
The last paragraph in the introduction, where the aims of the study are stated, needs to be more concise.
Line 75 – 79. The authors believe the objectives of the study have been stated clearly and concisely. Results of each objective were presented in the same order as stated in the last paragraph of the introduction.
Reviewer 2 Report
The study by Yuen et al reports the results of a serosurvey conducted amongst horses in Queensland, Australia during 2018-2020. The results inform on the apparent prevalence of four alphaviruses as measured by neutralizing antibody titer. Conclusions include the relative occurrence of these viruses by region and season, as well as apparent sera cross-reactivity between Ross River virus and Getah virus. Overall this is a useful contribution and should be valuable resource for public health epidemiology in Australia.
Specific comments:
1. Line 43, to avoid any possible confusion, please specify that you are specifically referring to Australia in this sentence. “Generally, the main reservoir hosts for arthropod-borne alphaviruses in Australia are marsupials.”
2. Please explain what the different categories in Supp Fig 1A represent (high-high, high-low, etc.).
3. Some of the figures are a bit difficult to interpret. In 2A and 3A, some of the symbols (dots) are extremely small and could easily be overlooked. In 2B, the northernmost district appears to be an off-white shade that is intermediate between the 0-20% category and the >20-40% categories.
Author Response
The study by Yuen et al reports the results of a serosurvey conducted amongst horses in Queensland, Australia during 2018-2020. The results inform on the apparent prevalence of four alphaviruses as measured by neutralizing antibody titer. Conclusions include the relative occurrence of these viruses by region and season, as well as apparent sera cross-reactivity between Ross River virus and Getah virus. Overall this is a useful contribution and should be valuable resource for public health epidemiology in Australia.
We thank the reviewer for the positive comments.
Specific comments:
- Line 43, to avoid any possible confusion, please specify that you are specifically referring to Australia in this sentence. “Generally, the main reservoir hosts for arthropod-borne alphaviruses in Australia are marsupials.”
Line 43 – Changes made as suggested.
- Please explain what the different categories in Supp Fig 1A represent (high-high, high-low, etc.).
A statement has been added in Supp Fig 1A.
- Some of the figures are a bit difficult to interpret. In 2A and 3A, some of the symbols (dots) are extremely small and could easily be overlooked. In 2B, the northernmost district appears to be an off-white shade that is intermediate between the 0-20% category and the >20-40% categories.
Fig 2A, 3A – The size of the symbols has been optimised. Size of the symbols is proportional to the sample size in the respective SLA3, and is generated automatically by the software (QGIS). If the symbol size with small number of samples were to increase, symbol size of areas with large sample number would therefore also increase. This would cause an overlay of the symbols to other SLA3, especially in SE QLD. The exact sample number for each SLA3 have been recorded in detail in Table 4.
Fig 2B – Again, the color scale is generated automatically by the software (QGIS). It is acknowledged that the 0-20% category is a very pale blue, but to the authors’ eyes it is clearly distinguishable from the white areas (i.e., areas not included in the analysis).
Reviewer 3 Report
In the study "Epidemiological Study of Multiple Zoonotic Mosquito-borne Alphaviruses in Horses in Queensland, Australia (2018 – 2020)" the authors conducted an Alphaviruses seroprevalence study in horses in Australia. In summary, the study showed a high seroprevalence of RRV and a cross-reaction among RRV and GEVT. The study is well explained and discussed. I have only a few suggestions that can improve the study.
INTRODUCTION:
In the paragraph of the objectives, I would also include the seroprevalence determination of GETV, including also a short paragraph about this virus in the lines above, where the authors are describing the other arboviruses studied here.
METHODOLOGY:
Cell culture and virus stock production: if available, the authors could provide the GenBank accession number of the virus used in this work.
RESULTS:
Table 3: Please, add the meaning of each acronym as a legend on the table.
Lines 166-169: I couldn't find the M&M of this experiment
Author Response
In the study "Epidemiological Study of Multiple Zoonotic Mosquito-borne Alphaviruses in Horses in Queensland, Australia (2018 – 2020)" the authors conducted an Alphaviruses seroprevalence study in horses in Australia. In summary, the study showed a high seroprevalence of RRV and a cross-reaction among RRV and GEVT. The study is well explained and discussed. I have only a few suggestions that can improve the study.
We thank the reviewer for the positive comments.
INTRODUCTION:
In the paragraph of the objectives, I would also include the seroprevalence determination of GETV, including also a short paragraph about this virus in the lines above, where the authors are describing the other arboviruses studied here.
Line 69 – 74. A brief introduction of GETV has been included as suggested.
METHODOLOGY:
Cell culture and virus stock production: if available, the authors could provide the GenBank accession number of the virus used in this work.
Information not available as virus stocks were donated by various research groups (acknowledged). However, where available, references to the original publication have been cited.
RESULTS:
Table 3: Please, add the meaning of each acronym as a legend on the table.
Line 170 – 171. Changes made as suggested.
Lines 166-169: I couldn't find the M&M of this experiment
M&M related to the determination of neutralising antibody titre to GETV and RRV have been described in section 2.2 and 2.3. Statistical analysis was described in line 138 – 139.